# Analysis of ICAM-1 rs3093030, VCAM-1 rs3783605, and E-Selectin rs1805193 Polymorphisms in African Women Living with HIV and Preeclampsia

**DOI:** 10.3390/ijms251910860

**Published:** 2024-10-09

**Authors:** Samukelisiwe Sibiya, Zinhle Pretty Mlambo, Mbuso Herald Mthembu, Nompumelelo P. Mkhwanazi, Thajasvarie Naicker

**Affiliations:** 1HIV Pathogenesis Programme, Doris Duke Medical Research Institute, College of Health Sciences, University of KwaZulu-Natal, Durban 4041, South Africa; mkhwanazi@ukzn.ac.za; 2Optics & Imaging, Doris Duke Medical Research Institute, College of Health Sciences, University of KwaZulu-Natal, Durban 4041, South Africa; zinhlemlambo66@gmail.com (Z.P.M.); mbusomthembu7@gmail.com (M.H.M.)

**Keywords:** preeclampsia, adhesion markers, ICAM-1, VCAM-1, E-selectin, endothelial dysfunction, HIV infection, single nucleotide polymorphisms

## Abstract

Intercellular adhesion molecule-1 (ICAM-1), vascular adhesion molecule-1 (VCAM-1), and E-selectin are cell adhesion molecules that play a significant role in inflammation and are implicated in the pathophysiology of preeclampsia development and HIV infection. More specifically, the immune expression of ICAM-1, VCAM-1, and E-selectin within cyto- and syncytiotrophoblast cells are dysregulated in preeclampsia, indicating their role in defective placentation. This study investigates the associations of ICAM-1, VCAM-1, and E-selectin gene variants (rs3093030, rs3783605, and rs1805193, respectively) with preeclampsia comorbid with HIV infection in women of African ancestry. It also examines the susceptibility to preeclampsia development and the effect of highly active antiretroviral therapy (HAART). A total of 405 women were enrolled in this study. Out of these women, 204 were preeclamptic and 201 were normotensive. Clinical characteristics were maternal age, weight, blood pressure (systolic and diastolic), and gestational age. Whole blood was collected, DNA was extracted, and genotyping of the ICAM-1 (rs3093030 C>T), VCAM-1(rs3783605 A>G), and E-selectin (rs1805193 A>C) gene polymorphisms was performed. Comparisons were made using the Chi-squared test. Our results demonstrated that preeclamptic women exhibited a higher frequency of analyzed variants, in contrast to those with the duality of preeclampsia and HIV infection. Additionally, the C allele of the ICAM-1 (rs3093030 C>T) and G allele of the VCAM-1 (rs3783605 A>G) genes were found to have a greater role in the co-morbidity and may be considered as a risk factor for preeclampsia development in women of African ancestry. In contrast, the SNP of rs1805193 of the E-selectin gene indicated that A>C was only significantly associated with HIV infection and not with preeclampsia. These findings highlight a strong association of the rs3093030 SNP of the ICAM-1 gene and of the VCAM-1 rs3783605 gene with the development of preeclampsia, indicating their role in the defective trophoblast invasion of preeclampsia. Sub-group analysis further reveals an association of the AA genotype with late-onset preeclampsia, a less severe form of disease indicating differing genetic predispositions between early and late-onset forms.

## 1. Introduction

Globally, Human Immunodeficiency Virus (HIV) infection affects an astounding 39.9 million people [1,2]. The expanded availability of antiretroviral therapy (ART) has increased life expectancy [3]. Moreover, the administration of ART during pregnancy has significantly enhanced the well-being of mothers, with a concomitant decline in new cases of pediatric HIV infections worldwide [4,5]. Nonetheless, pregnant women receiving ART face a substantially greater chance of experiencing adverse pregnancy outcomes such as stillbirth, preterm delivery, and delivering small for gestational age infants compared to HIV-naïve women [6,7].

The global prevalence of preeclampsia is 2–8%. It is a pregnancy-specific hypertensive disorder and a leading cause of maternal and fetal morbidity and mortality worldwide [8]. Preeclampsia disproportionately affects African women compared to women from other racial or ethnic backgrounds [9]. Its incidence rate is 60 percent higher in Black compared to White women, with the former group being more likely to develop severe preeclampsia [10,11]. 

The cell adhesion molecules, viz., intercellular adhesion molecule-1 (ICAM-1), vascular adhesion molecule-1 (VCAM-1), and E-selectin, play a significant role in inflammation. It is widely accepted that ICAM-1, VCAM-1, and E-Selectin are dysregulated, thereby affecting trophoblast invasion and angiogenesis, the hallmarks of preeclampsia development [12]. Of note, HIV is a powerful pro-angiogenic agent; however, ART reconstitutes the immune system while also dysregulating angiogenesis [7]. Recent studies have implicated single-nucleotide polymorphisms (SNPs) as a potential genetic marker associated with the development and severity of preeclampsia [13,14,15]. Additionally, in the comorbidity of HIV infection and PE, the interplay between these SNPs and endothelial dysfunction is further exacerbated [16]. Several SNPs including rs3093030, rs3783605, and rs1805193 have attracted considerable attention due to their involvement in various biological processes related to vascular homeostasis, inflammation, and immune response [17].

ICAM-1 is a single-chain 76–110 kDa glycoprotein found on the surface of immune and endothelial cells [18]. It facilitates interactions both between cells and the extracellular matrix and hence is significantly involved in the development and progression of tumors, particularly by promoting cell invasion and the spread of metastases [19,20]. Also, elevated serum levels of soluble ICAM-1 are observed in patients with cardiovascular diseases [21]. Of note, rs3093030 represents a polymorphism of the ICAM gene.

Dysregulation of TGF-β1 has been implicated in the pathogenesis of preeclampsia; thus, the SNP of rs3093030 will influence its expression and activity, contributing to endothelial dysfunction and the abnormal placentation of preeclampsia [22].

VCAM-1 (CD106), a part of the immunoglobulin superfamily, is a glycoprotein embedded in the cell membrane [23]. It is expressed in various cell types including vascular endothelial cells, fibroblasts, tissue macrophages, chondrocytes, dendritic cells, thymic epithelial cells, and neural tissue pericytes [24]. Its primary binding partner is the integrin α4β1 (very late antigen 4, VLA-4), and its expression on endothelial cells is induced by inflammatory signals [25]. Polymorphisms in this gene, including rs3783605, have been associated with altered eNOS activity and reduced nitric oxide bioavailability, potentially contributing to endothelial dysfunction in preeclampsia [26]. The rs3783605 polymorphism plays a role in VCAM-1 gene expression [25,26,27]. Of note, TNFα induces the VCAM-1 promoter in endothelial cells via NFK-B cells.

Altered expression levels of TNF-α have been observed in preeclampsia, and a rs3783605 SNP may influence TNF-α production, contributing to the immune dysregulation and endothelial dysfunction of this disorder [28]. Multiple variations in the promoter segment of the VCAM-1 gene influence its expression levels, with many of these variants proposed to be linked to a spectrum of inflammatory conditions [29].

The E-selectin gene (SEL-E), belonging to the selectin superfamily of adhesion molecules, is significant in the development of thrombovascular diseases [30]. The increased E-selectins reflect endothelial injury in preeclampsia, possibly a protective response to inhibit endothelial injury [31].

In this study, we aim to investigate the interplay between SNPs (rs3783605, rs3093030, and rs1805193) in preeclamptic women of African ancestry, comorbid with HIV infection, by analysing the genotype and allelic profiles of normotensive pregnant and preeclamptic women with HIV versus those without HIV women. We seek to elucidate potential synergistic effects and identify novel genetic markers that may contribute to the understanding and management of preeclampsia co-morbid with HIV infection.

## 2. Results

### 2.1. Clinical Characteristics 

The 405 pregnant women in this study were split into two groups: normotensive (*n* = 201) and preeclamptic (*n* = 204). By gestational age, the preeclamptic group was further divided into late-onset preeclampsia (LOPE) (*n* = 102) and early-onset preeclampsia (EOPE) (*n* = 102). Patient demographics and statistical differences are presented in Table 1. Preeclamptic pregnant women had a substantially lower gestational age at delivery (<0.0001) than the normotensive pregnant (N) group. Preeclamptic patients had significantly higher diastolic and systolic blood pressure than the normotensive pregnant group (<0.0001). Additionally, a significant difference in maternal weight was noted between the normotensive pregnant (0.0012) and the preeclamptic group. The maternal age was considerably greater when comparing the preeclamptic group to the normotensive pregnant group (0.0068). 

### 2.2. Genotype and Allele Frequencies of SNPs rs3093030, rs783605, and rs1805193

The association of gene polymorphisms of ICAM-1, VCAM-1, and E-selectin (rs3093030, rs783605, and rs1805193, respectively) across the study population is tabulated in Table 2. Four genetic models were tested: co-dominant (equal effect of two alleles from a gene pair), dominant (alleles with the same phenotype regardless of whether the paired allele is identical or not), recessive (creates a phenotype only when the paired alleles are identical), and over-dominant (heterozygote has a greater effect compared to the homozygote). Allelic and genotypic comparisons of frequencies were calculated. For each of the three variations under investigation, the relationships between these four genetic models and preeclampsia were investigated (Table 3, Table 4 and Table 5).

### 2.3. Rs3093030

#### 2.3.1. Genotypic Frequencies across Study Groups

*HIV status irrespective of pregnancy type*—The rs3093030 genotype frequencies were TT 26 (12.80%), TC 44 (21.67%), and CC 133 (65.51%) in HIV-negative compared to TT 30 (14.85%), TC 36 (17.82%), and CC 136 (67.32%) in HIV-positive women.Pregnancy type irrespective of HIV status—The rs3093030 genotype frequencies were TT 20 (9.95%), TC 39 (19.40%), and CC 142 (70.64%) in normotensive pregnant and TT 36 (17.64%), TC 41 (20.09%), and CC 127 (62.25%) in preeclamptic women.

More specifically, these frequencies were TT 11 (10.78%), TC 21 (20.58%), and CC 70 (68.67%) in EOPE and TT 25 (24.50%), TC 20 (19.60%), and CC 57 (55.88%) in LOPE women, irrespective of HIV status. 

*Across study groups*—The genotype frequencies of rs3093030 in normotensive pregnant HIV-negative women were TT 10 (9.80%), TC 21 (20.58%), and CC 71 (69.60%) compared to TT 16 (15.84%), TC 23 (22.77%), and CC 62 (61.38%) in the preeclamptic HIV-negative group. The genotype frequencies of rs3093030 in normotensive pregnant HIV-positive women were TT 10 (10.10%), TC 18 (18.18%), and CC 71 (71.71%) compared to TT 20 (19.4%), TC 18 (17.47%), and CC 65 (63.10%) in the preeclamptic HIV-positive groups (Table 2).

#### 2.3.2. Allelic Frequencies across Study Groups

*HIV status irrespective of pregnancy type*—The allele frequencies T and C were 96 (23.64%) and 310 (76.35%) in HIV-negative and 69 (23.76%) compared to 308 (76.23%) in HIV-positive women (Table 2).*Pregnancy type irrespective of HIV status*—The allele frequencies T and C were 79 (19.65%) and 323 (80.34%) in normotensive pregnant compared to 113 (27.69%) and 295 (72.30%) in the preeclamptic group. Stratification of the preeclamptic group showed the allele frequencies of T and C were 43 (21.07%) and 161 (78.92%) in EOPE compared to 70 (34.31%) and 134 (65.68%) in the LOPE group (Table 2).*Across the groups*—The allele frequencies of T and C were 41 (20.09%) and 163 (79.90%) in normotensive pregnant HIV-negative versus 55 (27.22%) and 147 (72.77%) in preeclamptic HIV-negative women, respectively. The allele frequencies T and C were 38 (19.19%) and 160 (80.80%) in normotensive pregnant HIV-positive and 58 (28.15%) and 148 (71.84%) in preeclamptic HIV-positive women (Table 2).

#### 2.3.3. Correlations between the Study Groups

a.Normotensive HIV-Negative Pregnant Women vs. Preeclamptic HIV-Negative

There were no significant correlations found between the genotypic frequencies of TT vs. CC, TT vs. TC, and TC vs. CC. The allelic frequency association between T and C did not significantly differ between normotensive pregnant HIV-negative women and preeclamptic HIV-negative women. Dominant (TT vs. TC + CC), recessive (TT + TC vs. CC), and/or over-dominant alleles (TT + CC vs. TC) did not significantly correlate with normotensive HIV-negative pregnant women (Table 3).

b.Normotensive Pregnant HIV-Positive vs. Preeclamptic HIV-Positive

The genotypic frequencies of TT vs. CC and TC vs. CC in the two groups did not significantly correspond, according to the data. However, there was also a significant difference in the allelic frequency association between T and C (*p* = 0.0274). Table 3 indicates that there were no significant associations found for the dominant, recessive, and over-dominant alleles of TT vs. TC + CC, TT + TC vs. CC, and TT + CC vs. TC.

c.Normotensive Pregnant vs. Preeclamptic Groups Irrespective of HIV Status

The genotypic frequency association of rs3093030 co-dominant TT vs. CC showed a significant association between normotensive pregnant and the preeclamptic group (*p* = 0.0270). However, the genotypic frequency of TT vs. TC and TC vs. CC showed no significant association between normotensive pregnant compared to the preeclamptic group. 

The dominant allele showed a significant association between the normotensive and preeclamptic group [(TT vs. TC + CC; *p* = 0.0209). Meanwhile, recessive (TT + TC vs. CC) and/or over-dominant alleles (TT + CC vs. TC) showed no significant association. The allelic frequency association between T and C showed a significant difference between normotensive pregnant and the preeclamptic group (*p* = 0.0081) (Table 3).

d.Early-Onset Preeclampsia vs. Late-Onset Preeclampsia Groups Irrespective of HIV Status

The genotypic frequency association of gene rs3093030 co-dominant TT vs. CC showed a significant association with EOPE compared to LOPE women irrespective of HIV status (*p* = 0.0135). However, the genotypic frequency of TT vs. TC and TC vs. CC showed no significant association with the EOPE compared to the LOPE group. 

The dominant alleles showed a significant association between the EOPE vs. LOPE group TT vs. TC + CC (*p* = 0.0161). Recessive (TT + TC vs. CC) and/or over-dominant alleles (TT + CC vs. TC) showed no significant association between EOPE vs. LOPE group. The allelic frequency association of T compared to C showed a significant difference between EOPE and LOPE groups (*p* = 0.0030) (Table 3).

e.Normotensive vs. Early-Onset Preeclamptic Women Irrespective of HIV Status

When comparing normotensive pregnant women to the EOPE group, the genotypic frequency association of rs3093030 co-dominant TT vs. CC revealed no significant association. Additionally, there was no significant correlation found between the genotypic frequency and TC vs. CC and normotensive pregnant women as compared to EOPE women. Dominant (TT vs. TC + CC), recessive (TT + TC vs. CC), and/or over-dominant alleles (TT + CC vs. TC) did not significantly correlate. Furthermore, there was no discernible difference between the normotensive pregnant group and the EOPE group according to the allelic frequency association of T vs. C (Table 3).

f.Normotensive vs. Late-Onset Preeclampsia Women Irrespective of HIV Status

A significant association of normotensive pregnant women with the LOPE group was noted in the genotypic frequency association of rs3093030 co-dominant TT vs. CC (*p* = 0.0008) and TT vs. TC (*p* = 0.0303). No significant association was noted in TC vs. CC irrespective of HIV status.

Between the normotensive pregnant group and the LOPE group, dominant and recessive alleles demonstrated a significant correlation: TT vs. TC + CC (*p* = 0.0011); TT + TC vs. CC (*p* = 0.0147). However, the over-dominant alleles did not significantly differ. 

The allelic frequency association of T vs. C showed a significant difference between normotensive pregnant and the LOPE group (*p* = 0.0001) (Table 3).

g.HIV-Negative vs. HIV-Positive Women Irrespective of Pregnancy Type

The genotypic frequency relationship of rs3093030 co-dominant TT vs. CC, TT vs. TC, and TC vs. CC did not significantly differ. Furthermore, no significant correlation was seen between the HIV-negative and HIV-positive groups for dominant (TT vs. TC + CC), recessive (TT + TC vs. CC), and/or over-dominant alleles (TT + CC vs. TC). Furthermore, there was no discernible difference between the HIV-positive and HIV-negative groups in the allelic frequency relationship T vs. C (Table 3).

### 2.4. rs3783605

#### 2.4.1. Genotypic Frequencies across Study Groups

*HIV status irrespective of pregnancy type*—The rs3783605 genotype frequencies were GG 18 (8.86%), AG 30 (14.77%), and AA 155 (76.35%) in HIV-negative women compared to GG 25 (12.37%), AG 36 (17.82%), and AA 141 (69.80%) in HIV-positive women.*Pregnancy type irrespective of HIV status*—The rs3783605 genotype frequencies were GG 10 (4.975%), AG 31 (15.42%), and AA 160 (79.60%) in normotensive pregnant and GG 33 (16.17%), AG 35 (17.15%), and AA 136 (66.66%) in PE women, irrespective of HIV status. When stratified by pregnancy type, these frequencies were GG 13 (12.74%), AG 12 (11.76%), and AA 77 (75.49%) in EOPE and GG 20 (19.60%), AG 23 (22.54%), and AA 59 (57.84%) in LOPE women.*Across study groups*—The genotype frequencies of rs3783605 in normotensive pregnant HIV-negative women were GG 3 (2.941%), AG 15 (14.70%), and AA 84 (82.35%) compared to GG 15 (14.85%), AG 15 (14.85%), and AA 62 (70.29%) in the preeclamptic HIV-negative group. The genotype frequencies of rs3783605 in normotensive pregnant HIV-positive women were GG 7 (7.07%), AG 16 (16.16%), and AA 76 (76.76%) compared to GG 18 (17.47%), AG 20 (19.41%), and AA 65 (63.10%) in the preeclamptic HIV-positive groups (Table 2).

#### 2.4.2. Allelic Frequencies across Study Groups

*HIV status irrespective of pregnancy type*—The allele frequencies of G and A were 66 (16.25%) and 340 (83.74%) in HIV-negative and 86 (21.28%) compared to 318 (78.71%) in HIV-positive women (Table 2).

*Pregnancy type irrespective of HIV status*—The allele frequencies of G and A were 51 (12.68%) and 351 (87.31%) in normotensive pregnant and 101 (24.75%) and 307 (75.24%) in the PE groups, irrespective of HIV status (Table 2). The allele frequencies of G and A were 38 (18.62%) and 166 (81.37%) in EOPE and 63 (30.88%) and 144 (69.11%) in the LOPE group irrespective of HIV status (Table 2).

*Across the groups*—The allele frequencies of G and A were 21 (10.29%) and 183 (89.70%) in normotensive HIV-negative pregnant compared to 45 (22.27%) and 157 (77.72%) in preeclamptic HIV-negative women, respectively. The allele frequencies of G and A were 30 (15.15%) and 168 (84.84%) in normotensive pregnant HIV-positive and 56 (27.18%) and 150 (72.81%) in preeclamptic HIV-positive women (Table 2).

#### 2.4.3. Correlations between the Study Groups

a.Normotensive HIV-Negative Pregnant Women vs. Preeclamptic HIV-Negative

The genotypic frequencies of AA vs. GG (*p* = 0.0026) and GG vs. AG (*p* = 0.0312) showed significant associations, while AG vs. AA showed no significant associations. The allelic frequency association between A and G also showed a significant difference between normotensive HIV-negative compared to preeclamptic HIV-negative women (*p* = 0.0012). 

Similarly, dominant (GG vs. AG + AA) and recessive alleles (GG + AG vs. AA) had significant associations between normotensive pregnant HIV-negative women, *p* = 0.0029 and *p* = 0.0486, respectively. Notably, over-dominant alleles and GG + AA vs. AG showed no significant difference with an adjusted *p*-value greater than 0.05 (Table 4). 

b.Normotensive HIV-Positive vs. Preeclamptic HIV-Positive

The findings showed a correlation between the genotypic frequencies of AA vs. GG (*p* = 0.0272), while GG vs. AG and AG vs. AA were not significant. However, the allelic frequency association between A and G also showed a significant difference (*p* = 0.0083), as did the dominant allele GG vs. AG + AA (*p* = 0.0300).

The recessive (GG + AG vs. AA) and over-dominant alleles (GG + AA vs. AG) showed no significant associations (Table 4).

c.Normotensive Pregnant vs. Preeclamptic Groups Irrespective of HIV Status

The genotypic frequency association of rs3783605 co-dominant AA vs. GG (*p* = 0.0001) and GG vs. AG (*p* = 0.0154) showed a significant association between normotensive pregnant and the preeclamptic group, whilst AG vs. AA showed no significant association.

Dominant [GG vs. AG + AA (*p* = 0.0003)] and recessive (GG + AG vs. AA, *p* = 0.0036) alleles showed a significant association between the normotensive and preeclamptic group. Meanwhile, the over-dominant allele (GG + AA vs. AG) showed no significant. The allelic frequency association between G and A showed a significant difference between normotensive pregnant and preeclamptic groups (*p ≤* 0.0001) (Table 4).

d.EOPE vs. LOPE

The genotypic frequency association of rs3783605 co-dominant AG vs. AA showed a significant association with EOPE compared to LOPE women irrespective of HIV status (*p* = 0.0228). However, the genotypic frequency of AA vs. GG and GG vs. AG showed no significant association when comparing the EOPE vs. LOPE group. 

A significant correlation was observed between recessive alleles and the EOPE vs. LOPE group [GG + AG vs. AA (*p* = 0.0113)]. Dominant and/or over-dominant alleles did not show a significant correlation between the EOPE and LOPE groups. There was a significant difference between the EOPE and LOPE groups in the allelic frequency association of T compared to C (*p* = 0.0057). (Table 4).

e.Normotensive vs. EOPE Women Irrespective of HIV Status

The genotypic frequency association of rs3783605 co-dominant AA vs. GG (*p* = 0.000111), GG vs. AG (*p* = 0.0422), and AG vs. AA (*p* = 0.0310) all showed a significant association between normotensive pregnant and EOPE groups.

Dominant and recessive showed a significant association between normotensive pregnant women compared to the EOPE group [GG vs. AG + AA (*p* = 0.0001)], [GG + AG vs. AA; (*p* = 0.0001)]. Also, the allelic frequency association of G vs. A showed a significant difference between normotensive pregnant women and the EOPE group (*p* ≤ 0.0001), while the over-dominant allele (GG + AA vs. AG) was not significant (Table 4).

f.Normotensive vs. Late-Onset Preeclampsia Women Irrespective of HIV Status

A significant association of normotensive pregnant with the LOPE group was noted in the genotypic frequency association of rs3783605 co-dominant AA vs. GG (*p* = 0.0364) and GG vs. AG (*p* = 0.0332). No significance was noted in AG vs. AA, irrespective of HIV status.

Dominant allele showed a significant association between normotensive pregnant vs. LOPE groups [GG vs. AG + AA (*p* = 0.0212)]. Recessive and over-dominant alleles showed no significant association between normotensive pregnant vs. LOPE groups. The allelic frequency association T vs. C was also not significant.

g.HIV-Negative vs. HIV-Positive Women Irrespective of Pregnancy Type

The genotypic frequency relationship of rs3783605 co-dominant AA vs. GG, AG vs. GG, and AG vs. AA did not significantly differ. Additionally, there was no discernible difference between HIV-positive and HIV-negative groups in dominant, recessive, or over-dominant alleles. Furthermore, there was no discernible difference between the HIV-positive and HIV-negative groups in the allelic frequency relationship G vs. A. (Table 4). 

### 2.5. Rs1805193

#### 2.5.1. Genotypic Frequencies across Study Groups

*HIV status irrespective of pregnancy type*—The rs1805193 genotype frequencies were CC 106 (52.21%), AC 59 (29.06%), and AA 38 (18.71%) in HIV-negative pregnant women compared to CC 83 (41.08%), AC 85 (42.07%), and AA 34 (16.83%) in HIV-positive pregnant women.*Pregnancy type irrespective of HIV status*—The rs1805193 genotype frequencies were CC 95 (47,26%), AC 69 (34.32%), and AA 37 (18.40%) in normotensive pregnant and AA 96 (47.05%), AC 73 (35.78%), and AA 35 (17.15%) in PE women, irrespective of HIV status. When stratified by pregnacy type, these frequencies were CC 45 (44.11%), AC 38 (37.25%), and AA 19 (18.62%) in EOPE and CC 51 (50.00%), AC 35 (34.31%), and AA 16 (15.68%) in LOPE women (Table 2).*Across study groups*—The genotype frequencies of rs1805193 in normotensive HIV-negative pregnant women were CC 56 (54.90%), AC 27 (26.47%), and AA 19 (18.40%) compared to CC 41 (40.59%), AC 41 (40.59%), and AA 19 (18.81%) in the preeclamptic HIV-negative group. The genotype frequencies of rs1805193 in normotensive HIV-positive pregnant women were CC 39 (39.39%), AG 42 (42.42%), and AA 18 (18.18%) compared to GG 46 (44.66%), AC 41 (39.80%), and AA 16 (15.53%) in the preeclamptic HIV-positive groups (Table 2).

#### 2.5.2. Allelic Frequencies across Study Groups

*HIV status irrespective of pregnancy type*—The allele frequencies of C and A were 271 (66.74%) and 135 (33.25%) in HIV-negative pregnant women and 253 (62.62%) compared to 151 (37.37%) in HIV-positive pregnant women (Table 2).

*Pregnancy type irrespective of HIV status*—The allele frequencies of C and A were 259 (65.62%) and 143 (32.81%) in normotensive pregnant women and 265 (64.95%) and 143 (35.04%) in the PE group. The allele frequencies of C and A were 128 (62.74%) and 76 (37.25%) in EOPE and 137 (67.15%) and 67 (32.84%) in LOPE (Table 2).

*Across the groups*—The allele frequencies of C and A were 143 (35.04%) and 259 (65.62%) in normotensive pregnant HIV-negative and 79 (39.10%) and 123 (60.89%) in preeclamptic HIV-negative groups. The allele frequencies of C and A were 120 (60.60%) and 78 (39.39%) in normotensive pregnant HIV-positive and 133 (64.56%) and 73 (35.43%) in preeclamptic HIV-positive groups.

#### 2.5.3. Correlations between the Study Groups

a.Normotensive HIV-Negative Pregnant Women vs. Preeclamptic HIV-Negative

The genotypic frequencies of AA vs. CC, CC vs. AC, and AC vs. AA in the two groups do not significantly correspond, according to the results. No significant relationships were observed when dominant, recessive, and over-dominant alleles were considered. Furthermore, there was no discernible difference between the two groups in the allelic frequency association between A and C (Table 5).

b.Normotensive HIV-Positive vs. Preeclamptic HIV-Positive

The genotypic frequencies of CC vs. AC, AA vs. CC, and AC vs. AA in the two groups do not significantly correspond. Additionally, when comparing normotensive HIV-positive pregnant women with preeclamptic HIV-positive groups, no significant was observed for dominant, recessive, or over-dominant alleles. Furthermore, there was no discernible difference between the two groups in the allelic frequency association between A and C (Table 5). 

c.Normotensive Pregnant vs. Preeclamptic Groups Irrespective of HIV Status

The genotypic frequencies of AA vs. CC, CC vs. AC, and AC vs. AA in the two groups did not significantly associate, according to the results. Similarly, no significant correlations between normotensive and preeclamptic groups were observed when dominant, recessive, and over-dominant alleles were considered. Furthermore, there was no discernible difference between the two groups in the allelic frequency connection between A and C (Table 5). 

d.EOPE vs. LOPE Groups Irrespective of HIV Status

The result showed no significant association between the genotypic frequencies of AA vs. CC, CC vs. AC, and AC vs. AA in the two groups. Similarly, when considering dominant, recessive, and over-dominant alleles, no significant associations were found between normotensive and preeclamptic groups. Additionally, the allelic frequency association between A and C also showed no significant difference between the two groups (Table 5). 

e.Normotensive vs. Early-Onset Preeclamptic Women Irrespective of HIV Status

The result showed no significant association between the genotypic frequencies of AA vs. CC, CC vs. AC, and AC vs. AA in the two groups. Similarly, when considering dominant, recessive, and over-dominant alleles, no significant associations were found between normotensive and preeclamptic groups. Additionally, the allelic frequency association between A and C also showed no significant difference between the two groups (Table 5).

f.Normotensive vs. LOPE Women Irrespective of HIV Status

The result showed no significant association between the genotypic frequencies of AA vs. CC, CC vs. AC, and AC vs. AA in the two groups. Similarly, when considering dominant, recessive, and over-dominant alleles, no significant associations were found between normotensive and preeclamptic groups. Additionally, the allelic frequency association between A and C also showed no significant (Table 5).

g.HIV-Negative vs. HIV-Positive Women Irrespective of Pregnancy Type

The genotypic frequency association of rs1805193 codominant, CC vs. AC (*p* = 0.0147), showed a significant association in HIV-negative compared to HIV-positive pregnant women. No other significant genotypic frequency was noted. Recessive [AA + AC vs. CC (*p* = 0.0466)] and over-dominant alleles [CC + AA vs. AC; (*p* = 0.0125)] showed a significant association with HIV-negative compared to HIV-positive pregnant women, irrespective of pregnancy type. Dominant alleles showed no significant association, and the allelic frequency association T vs. C was also not significant (Table 5).

## 3. Discussion

### 3.1. ICAM-1 rs3093030

In this study, normotensive pregnant HIV-negative, normotensive pregnant HIV-positive, preeclamptic HIV-positive, and preeclamptic HIV-negative women’s genotypic frequencies of rs3093030 variations (CC, CT, and TT) were investigated. With *p*-values of 0.2002, 0.2123, and 0.8559, respectively, the analysis demonstrated no significant associations of rs3093030 between normotensive pregnant HIV-negative women and preeclamptic HIV-negative women. The allelic frequency association between T and C did not significantly differ between normotensive pregnant HIV-negative women and preeclamptic HIV-negative women. Additionally, there was no significant correlation between the dominant, recessive, and over-dominant alleles and their respective ORs. The genotype frequencies analysis showed no significant associations between the genotypic frequencies of TT vs. CC, TT vs. TC, and TC vs. CC in the two groups of normotensive HIV-positive individuals compared to preeclamptic HIV-positive groups. Furthermore, there was a significant difference in the allelic frequency association between T and C. Furthermore, no significant relationships were found with the use of dominant, recessive, and over-dominant allele models in subsequent analyses. There were no significant associations found for the dominant, recessive, and over-dominant alleles of TT vs. TC + CC, TT + TC vs. CC, or TT + CC vs. TC.

These findings indicate that the rs3093030 polymorphism is significantly linked to a higher risk of preeclampsia development in HIV-positive pregnant women of African descent. Additionally, this study reveals that individuals with preeclampsia were more likely to have homozygous (CC and TT) genotypes, showing a heterozygous disadvantage [16]. The heterozygous genotype and TC alleles were more common in normotensive individuals compared to those with preeclampsia, regardless of HIV status. Statistical analysis of the dominant allele showed a significant association between the normotensive and preeclamptic groups, while recessive and over-dominant allele frequency associations showed no significant associations with preeclampsia.

A study has reported associations between rs3093030 variants and preeclampsia risk in different populations [32]. Zhang et al. (2016) found that rs3093030 CC genotype carriers have a high cancer susceptibility [33]. In contrast, Qui et al. showed that rs3093030 was not correlated with the susceptibility to colorectal cancer in Chinese patients [34]. The rs3093030 polymorphism is located at the non-coding exon region of the ICAM-1 gene and its variants may have the ability to regulate the expression of genetic information, thus influencing the function of ICAM-1 [22]. Also, the T allele of rs3093030 may be associated with an increased risk of preeclampsia development, possibly due to its influence on endothelial dysfunction and inflammation [32].

More recently, the upregulation of ICAM-1 was in response to protein tyrosine kinases proline-rich tyrosine kinase 2 (Pyk2) and focal adhesion kinase (FAK) [35]. Evidence regarding TNF-α has long indicated that, in the absence of other transcription factors, NF-κB is the main cause of ICAM-1 upregulation [36]. However, recent research indicates that although NF-κB causes an early increase in ICAM-1 and other inflammatory transcripts, the JAK/STAT pathway may be implicated by preserving the endothelium’s chronic inflammatory state once the cytokine stimulus has been eliminated [36].

Notably, all women living with HIV in our study received ART. There is limited research that specifically addresses the relationship between rs3093030 and ART response or outcomes. However, it is plausible that genetic variations within the ICAM-1 gene could influence the effectiveness of ART by affecting viral entry or immune responses [37]. Further studies are needed to elucidate the role of rs3093030 and other genetic variants in modulating ART response and HIV treatment outcomes.

### 3.2. VCAM-1 rs3783605

This study examined the genotypic frequencies of rs3783605 variants (AA, AG, and GG) in normotensive pregnant HIV-negative, normotensive pregnant HIV-positive, preeclamptic HIV-positive, and preeclamptic HIV-negative women. We report that in normotensive HIV-negative compared to preeclamptic HIV-negative, there is a significant association in the genotypic frequencies of AA vs. GG, while AG vs. AA demonstrated no significant associations with an adjusted *p* = 0.6939. The allelic frequency association between A and G showed a significant difference between normotensive HIV-negative compared to preeclamptic HIV-negative women. Similarly, dominant and recessive alleles (GG vs. AG + AA; GG + AG vs. AA) showed significant associations between normotensive pregnant HIV-negative women. Over-dominant alleles GG + AA vs. AG showed no significant difference with an adjusted *p*-value greater than 0.05. In the normotensive HIV-positive compared to preeclamptic HIV-positive groups, the genotype frequency analysis indicated a significant association between the genotypic frequencies of AA vs. GG, while GG vs. AG and AG vs. AA were not significant. However, the allelic frequency association between A and G and the dominant allele GG vs. AG + AA were significantly different. Meanwhile, recessive and over-dominant alleles of GG + AG vs. AA and GG + AA vs. AG showed no significant associations. 

The findings from our study indicate a significant association between the rs3783605 polymorphism and an increased risk of preeclampsia in HIV-positive pregnant women of African descent. We report that normotensive pregnant women had a higher frequency of the AA genotype compared to preeclamptic group and that the A allele of the VCAM-1 (rs3783605 A>G) single-nucleotide polymorphism plays a greater role in the pathogenesis of PE than the G allele and thus may be considered as a risk factor for PE development in an African population. One of the phenotypic features of endothelial dysfunction is the upregulation of cellular adhesion molecules (ICAM-1 and VCAM-1) and an increase in the expression of intercellular monocyte adherence to the endothelium [38]. Reactive oxygen species (ROS) production has been found to increase endothelial dysfunction, which has been linked to the development of preeclampsia [39]. 

In our study, the strong association of AA in preeclamptic vs. normotensive pregnancies may be due to the elevated VCAM-1 levels present during pregnancy and preeclampsia, implying a possible mechanism by which endothelial cells attract leukocytes and cause endothelial cell damage [40]. Research has also demonstrated that polymorphisms in the ICAM-1 and VCAM1 genes are associated with an increased genetic risk for various autoimmune diseases [41,42]. A study showed that the SNP of rs3783605 occurs at the Proto-Oncogene 2 Transcription Factor (ETS2) binding site and, as a result, could have a role in the etiology of diseases linked to VCAM-1 dysregulation such as multiple sclerosis, asthma, atherosclerotic lesions, thromboembolic disorders, multiple myeloma, insulin-dependent diabetes mellitus, and breast cancer [25]. 

In contrast to our findings, a comparison of genotypes between EOPE and LOPE patients has revealed that LOPE has a significantly higher frequency of the AA genotype in the VCAM-1 gene (*p* < 0.05) [32]. VCAM-1 is thought to be a useful marker for tracking leukocyte and endothelial activity [43]. It has been shown that the progression of a disease state correlates with VCAM-1 levels [44]. African Americans have many rare, physiologically active variations of the VCAM-1 promoter that may have an impact on the course of disease [45]. While the analysis of allele impact, including genotypes containing the G allele, showed no significant association in the preeclamptic group compared with control in Iraqi patients, VCAM-1 polymorphism analysis revealed a higher frequency of the A/G genotype in the control group compared to the preeclamptic group (100% vs. 88.2%) [46]. Therefore, based on these data, we can infer that rs3783605 A>G may have an impact on both the severity and course of VCAM-1-linked disease.

### 3.3. E-Selectin rs1805193

In this study, the genotypic frequencies of rs1805193 variations (AA, AC, and CC) were studied in normotensive pregnant HIV-negative, normotensive pregnant HIV-positive, preeclamptic HIV-positive, and preeclamptic HIV-negative women. However, there were no noteworthy correlations found in the analysis between normotensive HIV-negative pregnant women and preeclamptic HIV-negative women.

The allelic frequency association between A and C alleles showed no significant disparity between normotensive HIV-negative and preeclampsia HIV-negative groups. Dominant, recessive, and over-dominant allele models (AA vs. AC + CC; AA + AC vs. CC; and CC + AA vs. AC) also did not show a significant association. Genotype frequency analysis shows no significant associations between genotypic frequencies in normotensive HIV-positive compared to preeclamptic HIV-positive groups for CC vs AC and AC vs AA. Furthermore, there was no discernible change in the allelic frequency association between A and C. Also, no significant relationships were found by employing dominant, recessive, and over-dominant allele models in subsequent analysis.

AA vs. AC + CC, AA + AC vs. CC, and CC + AA vs. AC showed no significant differences. Previous studies have identified rs1805193 as a significant polymorphism associated with various cardiovascular conditions, including subclinical atherosclerosis in a Mexican and Iranian population [47,48]. The SNP rs1805193 is part of the E-selectin gene, which plays a crucial role in inflammation and endothelial function [48]. Since preeclampsia is characterized by endothelial dysfunction and inflammation, the genetic predisposition indicated by rs1805193 may contribute to the risk of developing PE [49]. Of note, Biwer et al. (2024) also reported that women with preeclampsia are predisposed to atherosclerosis [50]. HIV infection and preeclampsia are both associated with endothelial dysfunction and inflammation, which may explain the significant differences observed based on HIV status [51]. This is the first study to evaluate rs1805193 in a preeclamptic African population. Genetic predisposition may play a role, and lifestyle factors such as diet, physical activity, and smoking may all collaboratively impact preeclampsia development. Maintaining a healthy lifestyle may help mitigate the risks associated with genetic variation of rs1805193. The findings regarding the absence of association of rs1805193 with preeclampsia in South African women emphasizes the importance of considering genetic and ethnic differences when assessing preeclamptic risk.

Our findings suggest that the rs1805193 polymorphism is not significantly associated with an increased risk of preeclampsia development in HIV-positive pregnant women of African ancestry. Statistical analysis of the dominant, recessive, and over-dominant allele frequencies also showed no significant associations with preeclampsia.

Previous studies have also reported elevated maternal plasma E-selectin concentrations in women with preeclampsia at the time of delivery [52,53]. Despite being a derived allele, the C allele in the rs1805193 A>C SNP was found to be more common in our study than the ancestral A allele. This result supports the hypothesis that low-frequency (i.e., minor) derived variations account for the majority of hazardous alleles. The current conclusion, however, is in line with previous research that showed rare, derived alleles have a higher proportion of risk variations than ancestral alleles. Most late-onset disorders are brought on by neutral mutations, which can entirely replace ancestral alleles. This results in a situation where the derived (common) allele is the risk allele, and the ancestral (rare) allele is the protective one.

Also, Wei et al. (2011) demonstrated that genetic influences stemming from ancestral continent-of-origin impact endothelial cell pathology in African Americans [54]. 

The sample size of the study population was small, hence the frequencies of some homozygous variants were low. Additionally, the study groups were not age-matched. Moreover, only one polymorphism per adhesion molecule was studied, hence our finding of only one positive association. Analysis of rs3093030 of the ICAM-1 gene, rs378605 of the VCAM-1 gene, and rs1805193 of the E-selectin gene will have downstream effects on placentation as they are all cellular adhesion molecules. It is widely accepted that there is deficient trophoblast cell invasion and spiral artery remodeling in preeclampsia [55]. Since cell invasion requires proteolysis of the extracellular matrix, dysregulation of the cell adhesion molecules ICAM-1, VCAM-1, and E-selectin are implicated in preeclampsia development.

We also observed that women with preeclampsia had a lower gestational age at delivery compared to normotensive women. The significantly elevated systolic and diastolic blood pressure values in the preeclampsia group indicate the condition’s severity [56]. Additionally, there was a significant difference in maternal weight between the preeclampsia and normotensive groups, which has implications for the management of preeclampsia.

## 4. Methods and Materials

### 4.1. Study Population and Study Design

Four hundred and five pregnant women (*n* = 405) attending an antenatal clinic at a large regional hospital in eThekwini, South Africa, were recruited into this study. Institutional ethics approval (BREC/00002567/2021) of the retrospectively collected samples (BCA 338/17) was obtained and written informed consent were obtained from all women in the primary study. 

The sample size of the study population was determined by an institutional biostatistician using the Cohen effect [57]. The study population consisted of 405 pregnant women: PE (*n* = 204) and normotensive pregnant (*n* = 201) women. The PE group was equally divided into EOPE (*n* = 102) and LOPE (*n* = 102) and was further stratified by HIV status into EOPE− (*n* = 50), EOPE+ (*n* = 52), LOPE− (*n* = 51), and LOPE+ (*n* = 51). The normotensive group was divided by HIV status into N− (*n* = 102) and N+ (*n* = 99).

Preeclampsia was defined as having at least one positive proteinuria test obtained from a urine dipstick and a new-onset blood pressure measurement of ≥140/90 mmHg obtained twice, separated by four hours [58]. Late-onset preeclampsia (LOPE) was defined as the emergence of clinical signs at 34 weeks of gestation, whereas early-onset preeclampsia (EOPE) was defined as the manifestation of clinical indications before 33 weeks and 6 days of gestation [58].

Inclusion criteria

The study group consisted of primigravid and multigravida participants, diagnosed with PE (≥140/90 mmHg and the presence of a single incidence of proteinuria). Normotensive pregnant women served as the control group. The HIV status of the mother was determined by a rapid test. 

All HIV-positive trial participants received ART and prevention of mother-to-child transmission (PMTCT) therapy during pregnancy and lactation, regardless of their CD4 cell count. After nursing, women with CD4 counts < 500 cells/mm^3^ were also given continuous antiretroviral therapy. A single drug, such as zidovudine, also known as azido-thymidine (AZT), or a combination of several drugs, such as efavirenz (EFV), tenofovir disoprovil fumarate (TDF, Viread), and emtricitabine (FTC, Emtriva), were administered to women undergoing ARV treatment. Several individuals received an alternative medication combination consisting of Abacavar (ABC, Ziagen), Lamivudine (3TC, Epivir), and Efavirenz (EFV) in addition to PMTCT (nevirapine), in compliance with South African National HIV recommendations. Neonates exposed to HIV received nevirapine prophylaxis for four to six weeks.

Exclusion criteria

Women with polycystic ovarian syndrome, intrauterine death, cardiac illness, chorioamnionitis, unknown HIV status, eclampsia, sickle cell disease, systemic lupus erythematosus, pre-existing seizure disorders, thyroid disease, abruptio placentae, chronic renal disease, and patients who had been declined for participation were excluded. 

### 4.2. Blood Collection and DNA Isolation 

During the antepartum phase, peripheral venous blood was extracted and placed in EDTA anticoagulant tubes. Following the manufacturer’s instructions, DNA was extracted from whole blood using the QIAamp DNA Mini Kit (QIAGEN Sciences, Germantown, MD, USA). Samples were kept at −20 °C until the genotyping analysis was performed.

### 4.3. TaqMan Genotyping of Gene Polymorphisms

Using a TaqMan pre-designed SNP genotyping assay, three SNPs, Reference SNP cluster ID rs3783605, rs3093030, and rs1805193, were genotyped by the manufacturer’s protocol (Applied Biosystems by ThermoFisher Scientific, Foster City, CA, USA). 

With a total amount of 5.75 µL per well, the final reaction mixture contained DNA (3 µL), 2× TaqMan universal master mix (2.5 µL), and a 20× working stock of the TaqMan SNP genotyping assay (0.25 µL). Two PCR primers were used in the TaqMan genotyping experiment to amplify the target sequence and two labeled probes were used to find alleles. These probes distinguish between homogeneous and heterogeneous samples using two distinct fluorescent reporters—VIC and FAM dye. After PCR amplification, QuantStudioTM design and analysis software v2.5.1 were used to analyze genotyping and allelic discrimination data. 

### 4.4. Statistical Analysis 

The statistical program GraphPad Prism 5 (GraphPad Software, San Diego, CA, USA) was used to conduct the statistical analysis. The Mann–Whitney test was used to ascertain the statistical significance between the clinical features. Using the Hardy–Weinberg equilibrium test, the genotype conformance of the control and case groups was evaluated. Additionally, the Chi-square test was used to compare the subgroups. Using odds ratios (ORs) with 95% confidence intervals (CIs), the strength of the link was measured and presented. A strict standard for statistical significance was used, with statistical significance being defined as a *p*-value threshold of less than 0.05.

The Hardy–Weinberg equilibrium (HWE) test was employed to assess adherence to the observed genotype frequencies. Genotype presence was described using frequency and percentage. Subgroups were compared using either the Chi-squared test or Fisher’s exact test as appropriate. The strength of association was measured using odds ratios (ORs) alongside their respective 95% confidence intervals (CIs) for categorical data, while numerical data were assessed using Wilcoxon rank sum tests. A significance threshold of *p* < 0.05 was applied. Demographic analyses were conducted using one-way ANOVA tests with GraphPad Prism 8.43 software (GraphPad Software, San Diego, CA, USA), and multiple comparisons were adjusted using the Bonferroni correction test.

## 5. Conclusions

In conclusion, this study shows that the rs3093030 and rs378605 gene polymorphisms are risk factors for preeclampsia development in women of African ancestry. Of note, both single-nucleotide polymorphisms affect the severity as well as the progression of HIV infection. In summary, while rs3093030 of the ICAM-1 gene and rs3783605 of the VCAM-1 gene may play a role in HIV infection and disease progression, their exact impact remains uncertain and requires further investigation, particularly in the context of antiretroviral therapy. Single-nucleotide polymorphisms of ICAM-1, VCAM-1, and E-selectin will have downstream effects on their release into circulation, thus affecting placentation in preeclampsia. Deficient trophoblast cell invasion and lack of myometrial remodeling results in a hypoxic microenvironment with resultant adverse maternal and perinatal outcomes [59].

## Figures and Tables

**Table 1 ijms-25-10860-t001:** Women’s demographic and clinical data (*n* = 405).

Patient Data	N− (*n* = 102)	N+ (*n* = 99)	EOPE− (*n* = 50)	EOPE+ (*n* = 52)	LOPE− (*n* = 51)	LOPE+ (*n* = 51)	*p* Value
Maternal Age (years)	23.00 (20.00–28.00)	27.00 (23.00–32.00)	30.00 (23.25–35.00)	30.00 (27.00–33.00)	24.00 (20.75–30.00)	29.00 (24.00–32.00)	
N vs. EOPE							<0.0001 ****
N vs. LOPE							0.9639
EOPE vs. LOPE							0.0068 **
Systolic BP (mmHg)	119.0 (111.0–124.0)	114.0 (109.0–120.0)	161.0 (155.0–168.0)	161.0 (154.0–165.0)	159.0 (155.0–168.8)	155.0 (148.0–164.0)	
N vs. EOPE							<0.0001 ****
N vs. LOPE							<0.0001 ****
EOPE vs. LOPE							0.9162
Diastolic BP (mmHg)	71.00 (66.00–78.00)	71.00 (65.00–75.00)	104.0 (96.75–107.0)	104.0 (94.00–111.0)	101.5 (94.00–107.0)	99.00 (96.00–105.0)	
N vs. EOPE							<0.0001 ****
N vs. LOPE							<0.0001 ****
EOPE vs. LOPE							0.9543
Gestational Age (weeks)	39.00 (38.00–40.00)	38.00 (37.00–39.00)	30.00 (27.00–32.00)	29.00 (25.00–32.00)	38.00 (36.00–39.00)	37.00 (35.00–38.00)	
N vs. EOPE							<0.0001 ****
N vs. LOPE							<0.0001 ****
EOPE vs. LOPE							<0.0001 ****
Maternal Weight (Kg)	71.50 (60.15–83.63)	70.00 (62.80–80.00)	73.00 (63.93–90.00)	73.10 (65.00–89.70)	73.00 (63.50–89.40)	77.00 (68.00–101.0)	
N vs. EOPE							0.1317
N vs. LOPE							0.0012 **
EOPE vs. LOPE							0.5743

N−, normotensive HIV-negative; N+, normotensive HIV-positive; EOPE−, early-onset preeclampsia HIV-negative; EOPE+, early-onset preeclampsia HIV-positive; LOPE−, late-onset preeclampsia HIV-negative; LOPE+, late-onset preeclampsia HIV-positive. All values are represented as median (IQR). Asterisks (*) denote significance: ** *p* < 0.01 and **** *p* < 0.0001.

**Table 2 ijms-25-10860-t002:** Genotype and allele frequency distribution (%) gene polymorphisms (rs3093030, rs3783605, and rs1805193), respectively, of the control, preeclampsia, early-onset preeclampsia, late-onset preeclampsia, HIV-negative, and HIV-positive groups.

Genotype and Allelic Frequencies across Study Groups													
ICAM-1 Rs3093030 C>T		N− (*n* = 102)	N+ (*n* = 99)	EOPE (*n* = 102)	LOPE (*n* = 102)	EOPE− (*n* = 50)	EOPE+ (*n* = 52)	LOPE− (*n* = 51)	LOPE+ (*n* = 51)	N (N = 201)	PE (*n* = 204)	HIV− (*n* = 203)	HIV+ (*n* = 202)
Genotype Codominant	CC	71 (69.60%)	71 (71.71%)	70 (68.62%)	57 (55.88%)	35 (70.00%)	35 (67.30%)	27 (52.94%)	30 (58.82%)	142 (70.64%)	127 (62.25%)	133 (65.51%)	136 (67.32%)
	CT	21 (20.58%)	18 (18.18%)	21 (20.58%)	20 (19.60%)	10 (20.00%)	11 (21.15%)	13 (25.49%)	7 (13.72%)	39 (19.40%)	41 (20.09%)	44 (21.67%)	36 (17.82%)
	TT	10 (9.80%)	10 (10.10%)	11 (10.78%)	25 (24.50%)	5 (10.00%)	6 (11.53%)	11 (21.56%)	14 (27.45%)	20 (9.950%)	36 (17.64%)	26 (12.80%)	30 (14.85%)
Allele	C major	163 (79.90%)	160 (80.80%)	161 (78.92%)	134 (65.68%)	80 (80.00%)	81 (77.88%)	67 (65.68%)	67 (65.68%)	323 (80.34%)	295 (72.30%)	310 (76.35%)	308 (76.23%)
	T minor	41 (20.09%)	38 (19.19%)	43 (21.07%)	70 (34.31%)	20 (20.00%)	23 (22.11%)	35 (34.317%)	35 (34.31%)	79 (19.65%)	113 (27.69%)	96 (23.64%)	96 (23.76%)
VCAM-1 Rs3783605 A>G		N− (*n* = 102)	N+ (*n* = 99)	EOPE (*n* = 102)	LOPE (*n* = 102)	EOPE− (*n* = 50)	EOPE+ (*n* = 52)	LOPE− (*n* = 51)	LOPE+ (*n* = 51)	N (N = 201)	PE (*n* = 204)	HIV− (*n* = 203)	HIV+ (*n* = 202)
Genotype Codominant	AA	84 (82.35%)	76 (76.76%)	77 (75.49%)	59 (57.84%)	38 (76.00%)	39 (75.00%)	33 (64.70%)	26 (50.98%)	160 (79.60%)	136 (66.66%)	155 (76.35%)	141 (69.80%)
	AG	15 (14.70%)	16 (16.16%)	12 (11.76%)	23 (22.54%)	5 (10.00%)	7 (13.46%)	10 (19.60%)	13 (25.49%)	31 (15.42%)	35 (17.15%)	30 (14.77%)	36 (17.82%)
	GG	3 (2.941%)	7 (7.070%)	13 (12.74%)	20 (19.60%)	7 (14.00%)	6 (11.53%)	8 (15.68%)	12 (23.52%)	10 (4.975%)	33 (16.17%)	18 (8.866%)	25 (12.37%)
Allele	A major	183 (89.70%)	168 (84.84%)	166 (81.37%)	141 (69.11%)	81 (81.00%)	85 (81.73%)	76 (74.50%)	65 (63.72%)	351 (87.31%)	307 (75.24%)	340 (83.74%)	318 (78.71%)
	G minor	21 (10.29%)	30 (15.15%)	38 (18.62%)	63 (30.88%)	19 (19.00%)	19 (18.26%)	26 (25.49%)	37 (36.27%)	51 (12.68%)	101 (24.75%)	66 (16.25%)	86 (21.28%)
E-selectin Rs1805193 A>C		N− (*n* = 102)	N+ (*n* = 99)	EOPE (*n* = 102)	LOPE (*n* = 102)	EOPE− (*n* = 50)	EOPE+ (*n* = 52)	LOPE− (*n* = 51)	LOPE+ (*n* = 51)	N (N = 201)	PE (*n* = 204)	HIV− (*n* = 203)	HIV+ (*n* = 202)
Genotype Codominant	AA	19 (18.62%)	18 (18.18%)	19 (18.62%)	16 (15.68%)	9 (18.00%)	10 (19.23%)	10 (19.60%)	6 (11.76%)	37 (18.40%)	35 (17.15%)	38 (18.71%)	34 (16.83%)
	AC	27 (26.47%)	42 (42.42%)	38 (37.25%)	35 (34.31%)	16 (32.00%)	22 (42.30%)	25 (49.01%)	19 (37.25%)	69 (34.32%)	73 (35.78%)	59 (29.06%)	85 (42.07%)
	CC	56 (54.90%)	39 (39.39%)	45 (44.11%)	51 (50.00%)	25 (50.00%)	20 (38.46%)	16 (31.37%)	26 (50.98%)	95 (47.26%)	96 (47.05%)	106 (52.21%)	83 (41.08%)
Allele	A major	65 (31.86%)	78 (39.39%)	76 (37.25%)	67 (32.84%)	34 (34.00%)	42 (40.38%)	36 (35.29%)	31 (30.39%)	143 (32.81%)	143 (35.04%)	135 (33.25%)	151 (37.37%)
	C minor	139 (68.13%)	120 (60.60%)	128 (62.74%)	137 (67.15%)	66 (66.00%)	62 (59.61%)	66 (64.70%)	71 (69.60%)	259 (65.625%)	265 (64.95%)	271 (66.74%)	253 (62.62%)

**Table 3 ijms-25-10860-t003:** Genotypic and allelic associations of rs3093030 gene polymorphisms across study groups.

SNP Rs3093030 C>T Genotype		N− vs. PE− OR (95% CI), *p*-Value	N+ vs. PE OR (95% CI), *p*-Value	EOPE− vs. EOPE+ OR (95% CI), *p*-Value	LOPE− vs. LOPE+ OR (95% CI), *p*-Value	N vs. PE OR (95% CI), *p*-Value	HIV− vs. HIV+ OR (95% CI), *p*-Value	N vs. EOPE OR (95% CI), *p*-Value	N vs. LOPE OR (95% CI), *p*-Value	EOPE vs. LOPE OR (95% CI), *p*-Value
Codominant	TT vs. CC	1.832 (0.7749–4.333) *p* = 0.2002	2.013 (0.9425–4.298 *p* = 0.0822	1.200 (0.3349–4.300) *p* = 0.9649	1.145 (0.4449–2.949) *p* = 0.9672	2.013 (1.108–3.656) *p* = 0.0270 *	1.128 (0.6336–2.010) *p* = 0.7692	1.116 (0.5065–2.457) *p* = 0.8392	3.114 (1.604–6.047) *p*= 0.0008 ***	0.3583 (0.1625–0.7901) *p* = 0.0135 *
	TT vs. TC	0.5060 (0.1831–1.398) *p* = 0.2123	0.6327 (0.2589–1.546) *p* = 0.3770	0.9167 (0.2121–3.963) *p* ≥ 0.9999	0.4231 (0.1259–1.421) *p* = 0.2312	0.5840 (0.2898–1.177) *p* = 0.1605	0.7091 (0.3572–1.408) *p* = 0.3844	0.9790 (0.3952–2.426) *p* ≥ 0.9999	0.4103 (0.1847–0.9111) *p* = 0.0303 ***	2.386 (0.9348–6.092) *p* = 0.1042
	TC vs. CC	1.079 (0.5226–2.227) *p* = 0.8559	0.7853 (0.4200–1.468) *p* = 0.5335	0.9091 (0.3424–2.414) *p* ≥ 0.9999	2.063 (0.7178–5.932) *p* = 0.2022	0.8507 (0.5162–1.402) *p* = 0.6104	1.250 (0.7571–2.063) *p* =0.4450	0.9155 (0.5010–1.673) *p* = 0.7594	0.7827 (0.4208–1.456) *p* = 0.5171	1.170 (0.5777–2.368) *p* = 0.7220
Dominant	TT vs. TC + CC	0.5367 (0.2304–1.250) *p* = 0.2057	0.5367 (0.2396–1.067) *p* = 0.0883	0.8519 (0.2425–2.992) *p* > 0.9999	0.7268 (0.2932–1.801) *p* = 0.6458	0.4972 (0.2766–0.8938) *p* = 0.0209 *	0.8422 (0.4783–1.483) *p* = 0.5680	0.9141 (0.4199–1.990) *p* = 0.8424	0.3403 (0.1784–0.6492) *p* = 0.0011 *	2.686 (1.242–5.810) *p* = 0.0161 *
Recessive	TT + TC vs. CC	0.8407 (0.4128–1.712) *p* = 0.5451	0.6504 (0.3863–1.095) *p* = 0.1227	0.8824 (0.3818–2.039) *p* = 0.8327	1.270 (0.5802–2.779) *p* = 0.6903	0.6853 (0.4524–1.038) *p* = 0.0921	1.085 (0.7178–1.639) *p* = 0.7525	0.9089 (0.5424–1.524) *p* = 0.7910	0.5263 (0.3208–0.8634) *p* = 0.0147 *	1.727 (0.9741–3.062) *p* = 0.0827
over dominant	TT + CC vs. TC	0.8407 (0.4128–1.712) *p* = 0.7188	1.132 (0.6119–2.094) *p* = 0.7584	1.073 (0.4105–2.806) *p* ≥ 0.9999	0.4650 (0.1683–1.285) *p* = 0.2118	1.045 (0.6404–1.705) *p* = 0.9010	1.077 (0.5946–1.950) *p* = 0.8789	1.013 (0.5554–1.848) *p* > 0.9999	1.156 (0.5573–2.413) *p* = 0.7191	1.063 (0.5357–2.109) *p* > 0.9999
Allele (Major vs. minor)	T vs. C	1.382 (0.816–2.442) *p* = 0.1884	1.613 (1.065 – 2.442) *p* = 0.0274 *	1.136 (0.5787–2.229) *p* = 0.7343	1.000 (0.5609–1.783) *p* ≥ 0.9999	1.566 (1.128–2.174) *p* = 0.0081 **	1.006 (0.7280–1.392) *p* ≥ 0.9999	1.092 (0.7197–1.657) *p* = 0.6698	2.136 (1.461–3.122) *p* = 0.0001 ***	0.5113 (0.3281–0.7968) *p* = 0.0039 **

OR, odds ratio; CI, confidence intervals; N−, normotensive HIV-negative; N+, normotensive HIV-positive; PE−, preeclamptic HIV-negative; PE+, preeclamptic HIV-positive; EOPE−, early-onset preeclampsia HIV-negative; EOPE+, early-onset preeclampsia HIV-positive; LOPE−, late-onset preeclampsia HIV-negative; LOPE+, late-onset preeclampsia HIV-positive. Asterisks (*) denote significance: * *p* < 0.05, ** *p* < 0.01, and *** *p* < 0.001.

**Table 4 ijms-25-10860-t004:** Genotypic and allelic associations of rs3783605 gene polymorphisms across study groups.

SNP Rs3783605 A>G Genotype		N− vs. PE− OR (95% CI), *p*-Value	N+ vs. PE OR (95% CI), *p*-Value	EOPE− vs. EOPE+ OR (95% CI), *p*-Value	LOPE− vs. LOPE+ OR (95% CI), *p*-Value	N vs. PE OR (95% CI), *p*-Value	HIV− vs. HIV+ OR (95% CI), *p*-Value	N vs. EOPE OR (95% CI), *p*-Value	N vs. LOPE OR (95% CI), *p*-Value	EOPE vs. LOPE OR (95% CI), *p*-Value
Codominant	AA vs. GG	5.915 (1.645–2.127) *p* = 0.0026 **	2.634 (1.112–6.243) *p* = 0.0272 *	0.8352 (0.2570–2.714) *p* ≥ 0.9999	1.904 (0.6783–5.343) *p* = 0.3012	3.882 (1.845–8.167) *p* = 0.0001 ***	1.527 (0.7990–2.917) *p* = 0.2531	5.424 (2.399–12.26) *p* ≤ 0.0001 ***	2.701 (1.134 –6.437) *p* = 0.0364 *	2.008 (0.9237–4.364) *p* = 0.0832
	AG vs. GG	0.2000 (0.0477–0.8372) *p* = 0.0312 *	0.4640 (0.1694–1.271) *p* = 0.1515	1.633 (0.3353–7.957) *p* = 0.6951	0.8667 (0.2567–2.926) *p* ≥ 0.9999	0.3421 (0.1452–0.8062) *p* = 0.0154 *	0.8640 (0.3977–1.877) *p* = 0.8437	0.3710 (0.1462–0.9416) *p* = 0.0422 *	0.2978 (0.1031–0.8597) *p* = 0.0332 *	1.246 (0.4641–3.345) *p* = 0.8021
	AG vs. AA	1.183 (0.5410–2.587) *p* = 0.6939	1.222 (0.6351–2.353) *p* = 0.6251	1.364 (0.3980–4.675) *p* = 0.7594	1.650 (0.6244–4.360) *p* = 0.3358	1.328 (0.7780–2.268) *p* = 0.3404	1.319 (0.7721–2.254) *p* = 0.3418	2.012 (1.086–3.728) *p* = 0.0310 *	0.8044 (0.3916–1.652) *p* = 0.5982	2.501 (1.151–5.436) *p* = 0.0228 *
Dominant	GG vs. AG + AA	0.1737 (0.04864–0.6206) *p* = 0.0029 **	0.3943 (0.1678–0.9264) *p* = 0.0300 *	1.248 (0.3884–4.010) *p* = 0.7727	0.6047 (0.2237–1.634) *p* = 0.4550	0.2713 (0.1298–0.5670) *p* = 0.0003 ***	0.6889 (0.3632–1.307) *p* = 0.2636	0.2147 (0.09624–0.4788) *p* = 0.0001 ***	0.3584 (0.1514–0.8489) *p* = 0.0212 *	0.5989 (0.2800–1.281) *p* = 0.2537
Recessive	GG + AG vs. AA	0.5071 (0.2609–0.9856) *p* = 0.0486 *	0.5965 (0.3444 –1.033) *p* = 0.0829	0.9474 (0.3840–2.337) *p* ≥ 0.9999	0.5673 (0.2562–1.256) *p* = 0.2288	0.5051 (0.3223–0.7914) *p* = 0.0036 **	0.7158 (0.4602–1.113) *p* =0.1464	0.3516 (0.2087–0.5925) *p* = 0.0001 ***	0.7893 (0.4477–1.391) *p* = 0.4620	0.4455 (0.2449–0.8105) *p* = 0.0113 *
over dominant	GG + AA vs. AG	1.012 (0.4659 –2.197) *p* ≥ 0.9999	1.056 (0.5527 –2.016) *p* ≥ 0.9999	1.400 (0.4133 –4.742) *p* = 0.7605	1.403 (0.5505–3.574) *p* = 0.6362	1.116 (0.6582–1.892) *p* = 0.6894	1.251 (0.7366–2.123) *p* = 0.4226	1.597 (0.8745–2.915) *p* = 0.1526	0.7312 (0.7873–2.176) *p* = 0.4865	2.184 (1.020 –4.673) *p* = 0.0622
Allele (Major vs. minor)	A vs. G	2.498 (1.426–4.374) *p* = 0.0012 **	1.842 (1.176–2.886) *p* = 0.0083 **	0.9529 (0.4708–1.929) *p* ≥ 0.9999	1.664 (0.9121–3.035) *p* = 0.1293	2.264 (1.564–3.278) *p* ≤ 0.0001 ***	1.393 (0.9765–1.988) *p* = 0.0721	3.075 (2.025–4.670) *p* ≤ 0.0001 ***	1.575 (0.9957–2.493) *p* = 0.0531	1.952 (1.231–3.095) *p* = 0.0057 **

OR, odds ratio; CI, confidence intervals; N−, normotensive HIV-negative; N+, normotensive HIV-positive; PE−, preeclamptic HIV-negative; PE+, preeclamptic HIV-positive; EOPE−, early-onset preeclampsia HIV-negative; EOPE+, early-onset preeclampsia HIV-positive; LOPE−, late-onset preeclampsia HIV-negative; LOPE+, late-onset preeclampsia HIV-positive. Asterisks (*) denote significance: * *p* < 0.05, ** *p* < 0.01 and *** *p* < 0.001.

**Table 5 ijms-25-10860-t005:** Genotypic and allelic associations of rs1805193 gene polymorphisms across study groups.

SNP rs1805193 A>C Genotype		N− vs. PE− OR (95% CI), *p*-Value	N+ vs. PEOR (95% CI), *p*-Value	EOPE− vs. EOPE+ OR (95% CI), *p*-Value	LOPE− vs. LOPE+ OR (95% CI), *p*-Value	N vs. PE OR (95% CI), *p*-Value	HIV− vs. HIV+ OR (95% CI), *p*-Value	N vs. EOPE OR (95% CI), *p*-Value	N vs. LOPE OR (95% CI), *p*-Value	EOPE vs. LOPE OR (95% CI), *p*-Value
Codominant	AA vs. CC	0.8929 (0.4253–1.874) *p* = 0.8505	1.266 (0.6416–2.498) *p* = 0.4874	0.7200 (0.2455–2.111) *p* = 0.5928	1.733 (0.5479–5.484) *p* = 0.4004	1.068 (0.6211–1.838) *p* = 0.8902	0.8962 (0.5203–1.544) *p* = 0.7813	1.241 (0.6301–2.2446) *p* = 0.6120	0.9224 (0.4781–1.780) *p* = 0.8666	1.346 (0.6190–2.926) *p* = 0.5542
	CC vs. AC	1.327 (0.7008–2.514) *p* = 0.4193	0.7061 (0.4119–1.202) *p* = 0.2234	1.719 (0.7185–4.112) *p* = 0.2730	1.142 (0.4819–2.705) *p* = 0.8279	1.047 (0.6780–1.617) *p* = 0.9118	1.754 (1.131–2.722) *p* = 0.0147 *	0.9449 (0.5559–1.606) *p* = 0.8929	1.163 (0.6832–1.979) *p* = 0.5894	0.8127 (0.4417–1.495) *p* = 0.5370
	AC vs. AA	1.185 (0.5237–2.682) *p* = 0.8352	0.8939 (0.4512–1.771) *p* = 0.8627	1.238 (0.4089–3.745) *p* = 0.7809	1.979 (0.5894–6.646) *p* = 0.3678	1.118 (0.6341–1.973) *p* = 0.7727	1.572 (0.8886–2.782) *p* = 0.1462	1.173 (0.5745–2.395) *p* = 0.7208	1.072 (0.5431–2.118) *p* = 0.8645	1.094 (0.4874–2.455) *p* = 0.8402
Dominant	AA vs. AC + CC	0.9880 (0.4879–2.000) *p* ≥ 0.9999	1.073 (0.5730–2.009) *p* = 0.8723	0.9220 (0.3397–2.502) *p* ≥ 0.9999	1.829 (0.6106–5.481) *p* = 0.4148	1.089 (0.6543–1.814) *p* = 0.7954	1.138 (0.6832–1.896) *p* = 0.6969	1.213 (0.6381–2.304) *p* = 0.6326	0.9856 (0.5338–1.819) *p* ≥ 0.9999	1.094 (0.4874–2.455) *p* = 0.8402
Recessive	AA + AC vs. CC	0.8053 (0.4638–1.398) *p* = 0.4836	1.368 (0.8394–2.228) *p* = 0.2200	0.6250 (0.2844–1.373) *p* = 0.3188	1.082 (0.4975–2.351) *p* ≥ 0.9999	0.9918 (0.6713–1.465) *p* ≥ 0.9999	0.6648 (0.4491–0.9842) *p* = 0.0466 *	1.116 (0.6926–1.798) *p* = 0.7154	0.8809 (0.5456–1.422) *p* = 0.6274	1.267 (0.7301–2.198) *p* = 0.4832
over dominant	CC + AA vs. AC	1.288 (0.7015–2.366) *p* = 0.4425	0.7563 (0.4629–1.236) *p* = 0.3127	1.558 (0.6933–3.503) *p* = 0.3114	1.299 (0.5720–2.949) *p* = 0.6769	1.066 (0.7086–1.604) *p* = 0.8351	1.702 (1.127–2.572) *p* = 0.0125 *	0.994 (0.6049–1.651) *p* ≥ 0.9999	1.136 (0.6917–1.865) *p* = 0.6137	0.8798 (0.4960–1.560) *p* = 0.7703
Allele (Major vs. minor)	A vs. C	0.8818 (0.5833–1.333) *p* = 0.5987	1.205 (0.8486–1.710) *p* = 0.3226	0.7605 (0.4301–1.345) *p* = 0.3861	1.249 (0.6955–2.244) *p* = 0.5511	1.023 (0.7669–1.365) *p* = 0.8834	0.8347 (0.6254–1.114) *p* = 0.2396	1.129 (0.7904–1.613) *p* = 0.5281	0.9299 (0.6555–1.319) *p* = 0.7206	1.214 (0.8078–1.825) *p* = 0.4065

OR, odds ratio; CI, confidence intervals; N−, normotensive HIV-negative; N+, normotensive HIV-positive; PE−, preeclamptic HIV-negative; PE+, preeclamptic HIV-positive; EOPE−, early-onset preeclampsia HIV-negative; EOPE+, early-onset preeclampsia HIV-positive; LOPE−, late-onset preeclampsia HIV-negative; LOPE+, late-onset preeclampsia HIV-positive. Asterisks (*) denote significance: * *p* < 0.05.

## Data Availability

The data are unavailable due to privacy or ethical restrictions.

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
