# Peer review of "Analysis of ICAM-1 rs3093030, VCAM-1 rs3783605, and E-Selectin rs1805193 Polymorphisms in African Women Living with HIV and Preeclampsia"

_ijms, 2024, doi:10.3390/ijms251910860_

Round 1

Reviewer 1 Report

Comments and Suggestions for Authors

The article entitled “Analyzing ICAM-1 rs3093030, VCAM-1 rs3783605 and E-selectin rs1805193 Polymorphisms in African women living with HIV and preeclampsia” aims to assess the impact of certain polymorphisms on the development of preeclampsia in HIV positive and negative South African population. The main issues of this article are methodological.

My comments for the manuscript are:

Abstract

-              Abstract should be in a structured form

-              Replace HAART with highly active antiretroviral therapy (HAART)

Introduction

-              It is too thorough, and it can be shortened.

Results

-              The authors indicate that there statistically significant differences among study groups and control groups age. This fact means that PE group and N were not aged matched and furthermore inside the PE group the two subgroups were not aged matched. This is a major methodological flaw of the study

Discussion

-              The authors should also discuss the molecular mechanisms implicating in each polymorphism

-              The authors should discuss the effect of combinations of the aforementioned polymorphisms

-              The authors should also discuss the clinical implications of their finding and how it should be implemented in clinical practice

-              The limitations and strengths should be part of the discussion section

Methods

-              The authors mention “Sample of size of study population was not ideal hence the frequencies of some homozygous variants were low”. The calculation of the study sample should be discussed in the methods section.

Conclusions

-              Some sentences need rephrasing for better comprehensibility

Comments on the Quality of English Language

Extensive editing of English language required.

Author Response

Abstract

-              Abstract should be in a structured form

Response: We thank the reviewer for this comment, and we have edited the abstract accordingly.

-              Replace HAART with highly active antiretroviral therapy (HAART)

Response: Thank you for the feedback. We have replaced "HAART" with "highly active antiretroviral therapy (HAART)" in the abstract.

Introduction

-              It is too thorough, and it can be shortened.

 Response: Thank you for your input. We have removed an entire paragraph for conciseness.

Results

-              The authors indicate that there statistically significant differences among study groups and control groups age. This fact means that PE group and N were not aged matched and furthermore inside the PE group the two subgroups were not aged matched. This is a major methodological flaw of the study

Response: We acknowledge the concern regarding age matching between the study and control groups, as well as within the PE group subgroups. Nonetheless we have included this fact as a limitation of the study. The sample size will be drastically reduced if we had to aged match the samples. Please note that this study uses retrospectively collected samples.

Discussion

-              The authors should also discuss the molecular mechanisms implicating in each polymorphism

Response: Thank you for the suggestion. We have included the molecular mechanisms that maybe affected by the polymorphism associated with this gene in preeclampsia. It is outline in the discussion in (page 17, paragraph 4) (page 18, paragraph 2).

-              The authors should discuss the effect of combinations of the aforementioned polymorphisms

Response: Since the 3 SNPs emanate from cell adhesion molecules, analysis of rs3093030 of the ICAM-1 gene, rs378605 of the VCAM-1 gene and rs1805193 of the E-selectin gene will have downstream effect on placentation as they are all cellular adhesion molecules. It is widely accepted that there is deficient trophoblast cell invasion and spiral artery remodelling in preeclampsia. Since cell invasion requires proteolysis of the extracellular matrix hence ICAM-1, VCAM-1 and E-selectin will be affected in preeclampsia. (page 20, paragraph 2).

-              The authors should also discuss the clinical implications of their finding and how it should be implemented in clinical practice

Response: Single nucleotide polymorphism will have the downstream effect on the dysregulation of the soluble form of ICAM-1, VCAM-1 and E-selectin in circulation and effect placentation in preeclampsia. Deficient trophoblast cell invasion and lack of myometrial remodeling results in a hypoxic microenvironment with resultant with adverse maternal and perinatal outcome (page 22, paragraph 1).

-              The limitations and strengths should be part of the discussion section

Response: Thank you for your insightful comment. We will revise the discussion section to include both the limitations and strengths of our research. The limitation of the study includes the sample size which is included in the discussion (page 20). The strength of this study is that this is the first time the SNP of cell adhesion genes has been examined in women of African ancestry. Also note that this is the first study to examine the synergy of HIV infection and preeclampsia. Moreover, a unique feature of this study is the substantification of preeclampsia in EOPE and LOPE.

Methods

-              The authors mention “Sample of size of study population was not ideal hence the frequencies of some homozygous variants were low”. The calculation of the study sample should be discussed in the methods section.

Response: Thank you for pointing this out, the Cohen effect has been inserted in the methods. The sample size of the study population was determined by an institutional biostatistician using the Cohen effect. Sample size critically effect the reliability of results, particularly to demonstrate consistent genetic associations.

Conclusions

-              Some sentences need rephrasing for better comprehensibility

Response: Thank you for your feedback- Done.

Reviewer 2 Report

Comments and Suggestions for Authors

In this manuscript Sibiya et al evaluated ICAM-1 rs3093030, VCAM-1 rs3783605 and E-selectin rs1805193 polymorphisms in preeclamptic women, also including HIV positivity as a variable of interest.

Although the manuscript is interesting, the multiplicity of comparisons makes difficult to understand what exactly the authors mean to evaluate. For example, what is the meaning of compare HIV positive versus negative individuals if both groups include preeclamptic and non-preeclamptic, normotensive and hypertensive women, and so on? The manuscript needs a focus.

Also, several other points should be addressed:

It is quite important to revise the English-language usage and grammar. In fact, there are several points where it is not possible to follow the authors ideas (see for example, at the Abstract “…These findings highlight a strong genetic genotype (sic) of C>T of the ICAM-1 (rs3093030) gene and A>G of the VCAM-1rs3783605) gene in preeclampsia…”, or at Introduction “It is widely accepted that ICAM-1, VCAM-1 and E-Selectin are dysregulated…” why? When?).

When referring to “…elevated levels of ICAM-1…” are authors talking about soluble molecules or expression at cell surface. Please define.

It would be important to clearly state what is novel and which results just corroborate previous data. For example “Both systolic and diastolic blood pressure values were significantly higher in preeclampsia compared to the normotensive pregnant group respectively (<0.0001****).” Is not a novelty, since hypertension is one feature that defines preeclampsia.

Nothing is mentioned about medical regimens concerning the HIV positive individuals.

There is no mention about correction for multiple comparisons.

The number of individuals is quite small when all subgroupings are made. This limitation should be better highlighted at the Discussion.

Minor points

EOPE and LOPE are first defined at Table 1 legend, but these definitions should be explained at their first use at the text (RESULTS 1. Clinical characteristics).

Attention should be given to typos (for example, at Table 2 “FRUENCIES”)

Comments on the Quality of English Language

It is quite important to revise the English-language usage and grammar. In fact, there are several points where it is not possible to follow the authors ideas (see for example, at the Abstract “…These findings highlight a strong genetic genotype (sic) of C>T of the ICAM-1 (rs3093030) gene and A>G of the VCAM-1rs3783605) gene in preeclampsia…”, or at Introduction “It is widely accepted that ICAM-1, VCAM-1 and E-Selectin are dysregulated…” why? When?).

Attention should be given to typos (for example, at Table 2 “FRUENCIES”)

Author Response

Reviewer 2

Comments and Suggestions for Authors

In this manuscript Sibiya et al evaluated ICAM-1 rs3093030, VCAM-1 rs3783605 and E-selectin rs1805193 polymorphisms in preeclamptic women, also including HIV positivity as a variable of interest.

Although the manuscript is interesting, the multiplicity of comparisons makes difficult to understand what exactly the authors mean to evaluate. For example, what is the meaning of compare HIV positive versus negative individuals if both groups include preeclamptic and non-preeclamptic, normotensive and hypertensive women, and so on? The manuscript needs a focus.

Response: We thank you for this comment. Nonetheless, please note that the groups were selected due to high prevalence in SA. Moreover, they aid in illustrating the effect of HIV or the type of pregnancy on the effect of gene on associations.

Also, several other points should be addressed:

It is quite important to revise the English-language usage and grammar. In fact, there are several points where it is not possible to follow the authors ideas (see for example, at the Abstract “…These findings highlight a strong genetic genotype (sic) of C>T of the ICAM-1 (rs3093030) gene and A>G of the VCAM-1rs3783605) gene in preeclampsia…”, or at Introduction “It is widely accepted that ICAM-1, VCAM-1 and E-Selectin are dysregulated…” why? When?).

Response: Thank you for this comment, the paper has undergone grammar and spelling revisions.

Why- Preeclampsia has been previously associated with an imbalance of adhesion molecules in preeclampsia. In preeclampsia this has been associated with a shallow cytotrophoblast invasion and an absence of myometrial physiological conversion of spiral artery.

When- It is probable that the adhesion molecule dysregulation is associated with placentation which occurs early in gestation.

When referring to “…elevated levels of ICAM-1…” are authors talking about soluble molecules or expression at cell surface. Please define.

Response: The authors were referring to soluble forms of adhesion molecules present in the serum.

It would be important to clearly state what is novel and which results just corroborate previous data. For example, “Both systolic and diastolic blood pressure values were significantly higher in preeclampsia compared to the normotensive pregnant group respectively (<0.0001****).” Is not a novelty, since hypertension is one feature that defines preeclampsia.

Response: We apologize for the lack of clarity; preeclampsia is defined as a systolic blood pressure ≥ 140 mmHg or a diastolic blood pressure ≥ 90mmHg on two occasions at least 2 hours apart in previously normotensive patient.

Nothing is mentioned about medical regimens concerning the HIV positive individuals.

Response: This has been revised in the methods under the inclusion criteria.

There is no mention about correction for multiple comparisons.

Response: The authors have assessed the biological and practical significance of their results to ensure that they have meaningful implications.

The number of individuals is quite small when all subgroupings are made. This limitation should be better highlighted at the Discussion.

Response: Thank you for this comment. The sample size is now included under limitations of this study in the discussion.

Minor points

EOPE and LOPE are first defined at Table 1 legend, but these definitions should be explained at their first use at the text (RESULTS 1. Clinical characteristics).

Response: The text has been amended to reflect abbreviations at first in the text.

Attention should be given to typos (for example, at Table 2 “FRUENCIES”)

Response: Typos has been revised.

Reviewer 3 Report

Comments and Suggestions for Authors

Editing and English proofreading is needed: e.g.

The title must be improved, “analyzing” is not a proper word, maybe “analysis” would be better (???)

“.. that preeclamptic women exhibited a higher frequency of the C/T and A/G genotype” – They had a higher frequency of analyzed variants.

“ These findings highlight a strong genetic genotype of C>T of the ICAM-1 (rs3093030) gene…” – please clarify.

It seems that the authors have problems understanding the role of SNPs and gene-gene interactions as the following is not correct” The SNP rs3093030 of ICAM-1 is in the promoter region of the transforming growth factor-beta 1 (TGF-β1) gene,..”, “The SNP (rs3783605) of the VCAM-1 gene is in the promoter region of the endothelial nitric oxide synthase (eNOS) gene…”, “Rs3783605 is also found in the intron of the tumor necrosis factor-alpha (TNF-α) gene,”.

The increase in serum E-selectin levels during the acute phase of preeclampsia and the correlation between variations in the E-selectin gene and the occurrence of coronary artery lesions in adults imply a potential link between SELE variations and coronary artery disease in preeclampsia” – It is hard to accept this statement. Preeclampsia is related to hypertension but not to coronary artery disease. Please explain clearly.

Tables - When specific p-values are presented, asterisks are not necessary and only make the data harder to read by suggesting other relationships.

Tables 3, 4 and  5 - In the text of the manuscripts, please present only key associative links of potential practical importance. I suggest including the entire table as Supplementary Data.

The data description is very detailed. The text includes all the information contained in the tables. Such an arrangement does not add value to the manuscript. The description of the results should be modified. Attention should be paid to results that have potential biological and practical significance. Statistical significance does not accurately reflect the biological validity of the associations studied.

The discussion should not repeat the description of the test results. The discussion should include implications of the results, possible hypotheses, and their relationship to available data from other centers.

The conclusion is not correct. The presented study did not recognize that “rs3093030 and rs378605 gene polymorphism are major risk factors for preeclampsia development in women of African ancestry”. The results presented only show associations, but do not analyze the influence of other factors and the relationship between genetic variants and other factors related to the development of pre-eclampsia. Therefore, it cannot be concluded that these are the main risk factors. In addition, there is no need to conclude on the role of rs 1805193.

In conclusion, the manuscript needs significant improvement.

Comments on the Quality of English Language

needs improvement

Author Response

Reviewer 3

The title must be improved, “analyzing” is not a proper word, maybe “analysis” would be better (???)

Response: The title has been revised to reflect “analysis”.

“.. that preeclamptic women exhibited a higher frequency of the C/T and A/G genotype” – They had a higher frequency of analyzed variants.

Response: Done

“These findings highlight a strong genetic genotype of C>T of the ICAM-1 (rs3093030) gene…” – please clarify.

Response: These findings highlight a strong genetic association of rs3093030 SNP of the ICAM-1 gene in preeclampsia development (page 1, Abstract).

It seems that the authors have problems understanding the role of SNPs and gene-gene interactions as the following is not correct” The SNP rs3093030 of ICAM-1 is in the promoter region of the transforming growth factor-beta 1 (TGF-β1) gene,..”, “The SNP (rs3783605) of the VCAM-1 gene is in the promoter region of the endothelial nitric oxide synthase (eNOS) gene…”, “Rs3783605 is also found in the intron of the tumor necrosis factor-alpha (TNF-α) gene,”.

Response: These sentences are deleted from the introduction due to a request from the other 2 reviewers to make the introduction more concise.

” The increase in serum E-selectin levels during the acute phase of preeclampsia and the correlation between variations in the E-selectin gene and the occurrence of coronary artery lesions in adults imply a potential link between SELE variations and coronary artery disease in preeclampsia” – It is hard to accept this statement. Preeclampsia is related to hypertension but not to coronary artery disease. Please explain clearly.

Response: These sentences are deleted as well however we have inserted “The increased E-selectins reflect endothelial injury in preeclampsia possibly a protective response to inhibit the endothelial injury.”

Tables - When specific p-values are presented, asterisks are not necessary and only make the data harder to read by suggesting other relationships.

Response: Thank you- Done

Tables 3, 4 and 5 - In the text of the manuscripts, please present only key associative links of potential practical importance. I suggest including the entire table as Supplementary Data.

Response: With due respect, my supervisors has recommended that the tables remain as is, in line with the decision of the majority of reviewers.

The data description is very detailed. The text includes all the information contained in the tables. Such an arrangement does not add value to the manuscript. The description of the results should be modified. Attention should be paid to results that have potential biological and practical significance. Statistical significance does not accurately reflect the biological validity of the associations studied.

Response: The authors are conflicted about whether to remove descriptive information in the texts or to remove the tables. We feel more comfortable about retention of both as they help to clarify and illustrate the actual genetic associations to both the novice as well as molecular experts/molecular geneticist.

The discussion should not repeat the description of the test results. The discussion should include implications of the results, possible hypotheses, and their relationship to available data from other centers.

Response: The authors are conflicted about whether to remove descriptive information in the texts or to remove the tables. We feel more comfortable about retention of both as they help to clarify and illustrate the actual genetic associations to both the novice as well as molecular experts/molecular geneticist. Nonetheless the discussion now includes hypotheses and relationships from other studies.

The conclusion is not correct. The presented study did not recognize that “rs3093030 and rs378605 gene polymorphism are major risk factors for preeclampsia development in women of African ancestry”.

Response: Thank you, this comment the conclusion has been revised.

 The results presented only show associations, but do not analyze the influence of other factors and the relationship between genetic variants and other factors related to the development of pre-eclampsia.

Response: We are in agreement that our results do not analyze the influence of other factors in the development of preeclampsia, hence this sentence has been revised.

Therefore, it cannot be concluded that these are the main risk factors. In addition, there is no need to conclude on the role of rs 1805193.

Response: Thank you, this comment, this sentence has been deleted in the conclusion.

In conclusion, the manuscript needs significant improvement

Round 2

Reviewer 1 Report

Comments and Suggestions for Authors

The authors adequately replied to the majority of issues raised.

Comments on the Quality of English Language

Moderate editing of English language required.

Author Response

Open Review (x) I would not like to sign my review report
( ) I would like to sign my review report Quality of English Language ( ) I am not qualified to assess the quality of English in this paper.
( ) The English is very difficult to understand/incomprehensible.
( ) Extensive editing of English language required.
(x) Moderate editing of English language required.
( ) Minor editing of English language required.
( ) English language fine. No issues detected.            
  Yes Can be improved Must be improved Not applicable
Does the introduction provide sufficient background and include all relevant references? (x) ( ) ( ) ( )
Is the research design appropriate? ( ) ( ) (x) ( )
Are the methods adequately described? ( ) (x) ( ) ( )
Are the results clearly presented? ( ) (x) ( ) ( )
Are the conclusions supported by the results? (x) ( ) ( ) ( )
    Comments and Suggestions for Authors

The authors adequately replied to the majority of issues raised.

Comments on the Quality of English Language

Moderate editing of English language required.

Response: Thank you for your feedback. The manuscript has gone through language editing.

Submission Date 17 August 2024 Date of this review 08 Sep 2024 17:54:54 © 1996-2024 MDPI (Basel, Switzerland) unless otherwise stated

Reviewer 2 Report

Comments and Suggestions for Authors

Although authors approached the main questions raised by this reviewer, I am still concerned with the fact that the sample is quite fragmented when separated according to several features (i.e. HIV positivity, preeclampsia, early or late PE onset, etc). In this sense, the reading of the manuscript is quite confusing, and I strongly suggest to present separate Tables according to the feature evaluated.

Therefore, I believe that all te results section can be better presented.

Also, the discussion deserves a bit of attention.

Comments on the Quality of English Language

When restructuring the results section, attention should also be given to the English-language usage.

Author Response

Open Review (x) I would not like to sign my review report
( ) I would like to sign my review report Quality of English Language ( ) I am not qualified to assess the quality of English in this paper.
( ) The English is very difficult to understand/incomprehensible.
( ) Extensive editing of English language required.
( ) Moderate editing of English language required.
(x) Minor editing of English language required.
( ) English language fine. No issues detected.            
  Yes Can be improved Must be improved Not applicable
Does the introduction provide sufficient background and include all relevant references? (x) ( ) ( ) ( )
Is the research design appropriate? ( ) (x) ( ) ( )
Are the methods adequately described? ( ) (x) ( ) ( )
Are the results clearly presented? ( ) (x) ( ) ( )
Are the conclusions supported by the results? (x) ( ) ( ) ( )
    Comments and Suggestions for Authors

Although authors approached the main questions raised by this reviewer, I am still concerned with the fact that the sample is quite fragmented when separated according to several features (i.e. HIV positivity, preeclampsia, early or late PE onset, etc). In this sense, the reading of the manuscript is quite confusing, and I strongly suggest to present separate Tables according to the feature evaluated.

Therefore, I believe that all te results section can be better presented.

Response: Thank you for this comment. I understand your concerns about the fragmentation of the sample and the resulting confusion in the manuscript. To address this, we have decided to move the tables

Also, the discussion deserves a bit of attention.

Response: Thank you for your comment. We have edited the discussion section to ensure it is thoroughly addressed.

Comments on the Quality of English Language

When restructuring the results section, attention should also be given to the English-language usage.

Response: Thank you for your comment. The manuscript has gone through language editing.

Submission Date 17 August 2024 Date of this review 09 Sep 2024 16:50:29 © 1996-2024 MDPI (Basel, Switzerland) unless otherwise stated

Reviewer 3 Report

Comments and Suggestions for Authors

Page 2 – “The SNP rs3093030 of ICAM-1 is in the promoter region of transforming growth factor”.

I underlined this mistake before: The SNP of the ICAM gene is not a concurrent polymorphism in the TGF gene. ICAM and TGF are two different genes.  Two different rs therefore represent polymorphisms in these two genes.

Similarly: “Rs3783605 is also found in the intron of the tumor necrosis factor-alpha (TNF-α) gene”. Rs3783605 is the SNP of VCAM, not TNFα.

When we give p-values, we do not use asterisks (*) at the same time. You have to choose, either you give p<a, ablo *** and put the information what *** means in the legend.

I want to emphasize that, as a reviewer, I want authors to either try to conform to my comments or try to convince me of the correctness of their behavior. Citing a supervisor's opinion or claiming that the authors want to do it this way because it's their idea doesn't convince me. So I repeat my comments:

Tables 3,4,5 - The data description is very detailed. The text includes all the information contained in the tables. Such an arrangement does not add value to the manuscript. The description of the results should be modified. Attention should be paid to results that have potential biological and practical significance. Statistical significance does not accurately reflect the biological validity of the associations studied.

The discussion should not repeat the description of the test results. The discussion should include implications of the results, possible hypotheses, and their relationship to available data from other centers.

Comments on the Quality of English Language

Required improvements

Author Response

Open Review

Quality of English Language

( ) I am not qualified to assess the quality of English in this paper.
( ) The English is very difficult to understand/incomprehensible.
( ) Extensive editing of English language required.
(x) Moderate editing of English language required.
( ) Minor editing of English language required.
( ) English language fine. No issues detected.

  Yes Can be improved Must be improved Not applicable
Does the introduction provide sufficient background and include all relevant references? ( ) ( ) (x) ( )
Is the research design appropriate? (x) ( ) ( ) ( )
Are the methods adequately described? (x) ( ) ( ) ( )
Are the results clearly presented? ( ) ( ) (x) ( )
Are the conclusions supported by the results? ( ) (x) ( ) ( )

Comments and Suggestions for Authors

Page 2 – “The SNP rs3093030 of ICAM-1 is in the promoter region of transforming growth factor”.

I underlined this mistake before: The SNP of the ICAM gene is not a concurrent polymorphism in the TGF gene. ICAM and TGF are two different genes.  Two different rs therefore represent polymorphisms in these two genes.

Response: Thank you for this comment. We apologize for this error. Both ICAM and TGF are 2 different genes of which rs3093030 represents a variant of the former gene- revisions on page 2. 

Similarly: “Rs3783605 is also found in the intron of the tumor necrosis factor-alpha (TNF-α) gene”. Rs3783605 is the SNP of VCAM, not TNFα.

Response: Rs3783605 polymorphism plays a role in VCAM-1 gene expression. Off note, TNFα induces VCAM-1 promoter in endothelial cells via NFK-B cells. Revisions made in page 2.

When we give p-values, we do not use asterisks (*) at the same time. You have to choose, either you give p<a, ablo *** and put the information what *** means in the legend.

Response: Thank you. All the asterisks have been removed from text.

I want to emphasize that, as a reviewer, I want authors to either try to conform to my comments or try to convince me of the correctness of their behavior. Citing a supervisor's opinion or claiming that the authors want to do it this way because it's their idea doesn't convince me. So I repeat my comments:

Tables 3,4,5 - The data description is very detailed. The text includes all the information contained in the tables. Such an arrangement does not add value to the manuscript. The description of the results should be modified. Attention should be paid to results that have potential biological and practical significance. Statistical significance does not accurately reflect the biological validity of the associations studied.

Response: The manuscript has been revised and all tables has been moved into supplementary data.

The discussion should not repeat the description of the test results. The discussion should include implications of the results, possible hypotheses, and their relationship to available data from other centers.

Response: Thank you for this comment. We have revised the discussion and remove the description of the results.

Comments on the Quality of English Language

Required improvements

Submission Date

17 August 2024

Date of this review

08 Sep 2024 21:05:49

Round 3

Reviewer 3 Report

Comments and Suggestions for Authors

Abstract: please change "... a strong genetic association of rs3093030 SNP of the ICAM-1 gene and of the VCAM-1rs3783605" by removing "genetic".

I want to emphasize that I did not suggest moving all the tables with the results as supplemental data, but presenting an abbreviated version of the tables in the text, showing only the relevant elements, and then moving the tables with all the data as supplementary tables. In addition, I suggested shortening the description of the data and focusing on the essential elements. I was also inclined to leave the tables in a large version in the text, but necessarily shorten their description and focus on the key elements, not describing all the data in detail, because the interested reader will reach for it himself. I made this clear in the second review. Unfortunately, the authors chose the third option, in my opinion, the worst, they decided to move the tables and present them as appendices, leaving the whole description of the data. For the reader, this precise description is not as important as the data. Therefore, removing all the data from the manuscript makes no sense and will make no one interested in this publication. Therefore, I point out once again that the tables with the results must remain in the text (it can be in their expanded form, since, as I have noted, the creation of different versions of tables is not the preferred solution for the authors), and the description must be corrected and the statistically significant data must be referred to. A very detailed description of all data is unnecessary. 

-  

Comments on the Quality of English Language

minor editing required

Author Response

Abstract: please change "... a strong genetic association of rs3093030 SNP of the ICAM-1 gene and of the VCAM-1rs3783605" by removing "genetic".

Response: Thank you for this comment. Done.

I want to emphasize that I did not suggest moving all the tables with the results as supplemental data, but presenting an abbreviated version of the tables in the text, showing only the relevant elements, and then moving the tables with all the data as supplementary tables. In addition, I suggested shortening the description of the data and focusing on the essential elements. I was also inclined to leave the tables in a large version in the text, but necessarily shorten their description and focus on the key elements, not describing all the data in detail, because the interested reader will reach for it himself. I made this clear in the second review. Unfortunately, the authors chose the third option, in my opinion, the worst, they decided to move the tables and present them as appendices, leaving the whole description of the data. For the reader, this precise description is not as important as the data. Therefore, removing all the data from the manuscript makes no sense and will make no one interested in this publication. Therefore, I point out once again that the tables with the results must remain in the text (it can be in their expanded form, since, as I have noted, the creation of different versions of tables is not the preferred solution for the authors), and the description must be corrected and the statistically significant data must be referred to. A very detailed description of all data is unnecessary. 

Response: Thank you for this comment. We are in agreement that the tables represent the main part of the manuscript. Hence, we have now moved the tables back into the text. A more concise and shortened description of the results is now included.